# An Electrochemical Sensor Based on Gold-Nanocluster-Modified Graphene Screen-Printed Electrodes for the Detection of β-Lactoglobulin in Milk

**DOI:** 10.3390/s20143956

**Published:** 2020-07-16

**Authors:** Jingyi Hong, Yuxian Wang, Liying Zhu, Ling Jiang

**Affiliations:** 1College of Food Science and Light Industry, Nanjing Tech University, Nanjing 211816, China; hongjingyi1019@163.com; 2College of Biotechnology and Pharmaceutical Engineering, Nanjing Tech University, Nanjing 211816, China; yxwang@njtech.edu.cn; 3School of Chemistry and Molecular Engineering, Nanjing Tech University, Nanjing 211816, China; zlyhappy@njtech.edu.cn

**Keywords:** β-lactoglobulin, electrochemical sensors, PEI-rGO-AuNCs, screen-printed electrode

## Abstract

A simple and low-cost electrochemical sensor based on multimodified screen-printed electrodes (SPEs) was successfully synthesized for the sensitive detection of β-lactoglobulin (β-Lg). The surface treatment of SPEs was accomplished by a simple drip coating method using polyethyleneimine (PEI), reduced graphene oxide (rGO), and gold nanoclusters (AuNCs), and the treated SPEs showed excellent electrical conductivity. The modified SPEs were then characterized with UV-Vis, SEM, TEM, and FTIR to analyze the morphology and composition of the AuNCs and the rGO. An anti-β-Lg antibody was then immobilized on the composite material obtained by modifying rGO with PEI and AuNCs (PEI-rGO-AuNCs), leading to the remarkable reduction in conductivity of the SPEs due to the reaction between antigen and antibody. The sensor obtained using this novel approach enabled a limit of detection (LOD) of 0.08 ng/mL and a detection range from 0.01 to 100 ng/mL for β-Lg. Furthermore, pure milk samples from four milk brands were measured using electrochemical sensors, and the results were in excellent agreement with those from commercial enzyme-linked immunosorbent assay (ELISA) methods.

## 1. Introduction

The prevalence and severity of food allergies have increased significantly over the past few decades. The main food allergens are ubiquitous food proteins such as milk, eggs, wheat, fish, soy, peanuts, nuts, and shellfish [1]. Cow’s milk protein allergy (CMPA) refers to the most common food allergic disease, affecting between 2 and 3% of young children (0–2 years) [2]. CMPA can produce inflammatory factors and neurotoxic compounds, affect the development of the central nervous system, and gradually lead to functional disorders [3]. Milk protein allergic reactions involve multiple systems, including the skin, digestive tract, and respiratory tract; such reactions include allergic dermatitis and recurrent asthma, and the most serious reactions can result in death [4].

Whey protein is a protein in cow’s milk, including β-lactoglobulin (β-Lg), α-lactalbumin (α-La), serum albumin, immunoglobulin, and other trace proteins [5,6]. Among them, β-Lg is a very important protein in bovine whey protein, accounting for about 43.6–50.0% of the total bovine whey protein. It is reported that 82% of those affected by CMPA are allergic to β-Lg in the milk protein [7]. Therefore, β-Lg can be used as an effective indicator for detecting whether there is cow’s milk protein present in food components. Currently, there are many analytical methods for the detection of allergens, which can be divided into three categories according to detection principle: polymerase chain reaction (PCR) detection methods based on allergen DNA, chromatographic detection methods based on allergen proteins (e.g., HPLC [8,9], UPLC [10], and LC-MS [11,12]), and immunological detection methods (e.g., ELISA [13,14], ICA [15], and SPR [16,17,18]). Bonfatti et al. [8] developed a novel reversed-phase high-performance liquid chromatography (RP-HPLC) method that used a C8 column to separate and quantify the genetic variants of casein (CN) and whey protein, including α-La, β-Lg-A, β-Lg-B, α_S1_-casein, and α_S2_-casein. Boitz et al. [10] detected β-Lg in heat-treated milk samples by UPLC on a sub-2-μm particle reversed-phase column with a total runtime of 30 min. Ji et al. [11] established a method for confirming and quantifying milk allergens (α-La, β-Lg, and α_S1_-casein) in foods based on liquid chromatography–tandem mass spectrometry with multiple reaction monitoring (LC-MRM/MS). However, the above methods need expensive equipment and experienced laboratory technicians, and the preparation required prior to a test is complicated and takes a long time. In recent years, electrochemical sensors have been considered to be one of the preferred technologies due to their low cost, simple operation, portability, fast response speed, high sensitivity, and wide application range [19,20,21,22].

Graphene is the most commonly used sensing material for electrochemical sensors because of the enhancement of the specific surface area of the electrode and the interface electron transport speed [23]. In addition, graphene can improve the signal-to-noise ratio of the sensing signal, improving the sensitivity of sensing detection [24]. Compared with graphene, functionalizing graphene with polymer chains can make graphene easier to disperse in organic solvents and water. Many documents have described the covalent functionalization of graphene oxide (GO) with polymer chains [25,26,27,28]. The introduction of a charged soluble polymer chain unit onto the GO plane will produce a well-dispersed composite material. Besides, most metal nanomaterials have also been utilized in combination with GO due to their superior electrical conductivity. Metal nanoclusters have attracted much attention because of their extraordinary physical and chemical properties and have potential applications in optics, catalysts, sensing, targeted imaging, and therapy [29]. Gold nanoclusters (AuNCs), a class of particles less than 2 nm in diameter, have the advantages of good fluorescence performance and nontoxicity and are widely used in biosensing, bioimaging, cell labeling, drug delivery, and biomolecule (DNA, protein, enzyme) detection, etc. [30]. A composite material combining AuNCs and reduced GO (rGO) can greatly improve the dispersion of rGO. It will also produce “active sites” on the surface of rGO, and rGO provides an ideal carrier for the high activity of metal nanoparticles, which is conducive to improving its electrochemical activity [30,31].

In the existing methods, the sensor modification process is long and complicated. For example, the experiments of Eissa et al. [32] required multiple cyclic voltammetry scans to complete the modification of the sensor and the fixation of the antibody. In this study, we used self-assembly of rGO modified with polyethyleneimine (PEI) and AuNCs to obtain a composite material (PEI-rGO-AuNCs) for the surface modification of the working electrode of a set of screen-printed electrodes, which was successfully proved to detect the nonylphenols with a low detection limit [33]. The electrode surface can be modified by a simple drip coating method. An antibody can be fixed onto the modified electrode surface, and the current change caused by the reaction between antigen and antibody can be used to achieve the purpose of detection. In comparison to our previous strategy of using an ultra-highly selective DNA aptamer and choosing a complicated grafting protocol for the modified screen-printed carbon electrode [20,32], the major objective of this study is to establish an electrochemical sensor with simple operation, low cost, fast response, and moderate detection limit (meeting the requirements of the local government) that can be applied to portable field equipment for the rapid detection of β-Lg.

## 2. Experimental

### 2.1. Reagents and Materials

The anti-β-lactoglobulin antibody was obtained from Solarb (Beijing, China), and β-Lg was obtained from Sigma. The ELISA kit was purchased from Jiangsu Zeyu Biotechnology Co., Ltd. (Yancheng, China), and GO was purchased from Suzhou Tanfeng Graphene Technology Co., Ltd. (Suzhou, China).

### 2.2. Apparatus

The CHI 660E electrochemical workstation (Shanghai Chenhua Instrument Co., Ltd., China) was used to perform all the electrochemical measurements. Disposable stencil-printed electrodes (SPEs) were purchased from Kenslet (Zhejiang, China; reference number CL10). The sensor connector (Kenslet, Zhejiang, China) allowed the SPEs to be connected to an electrochemical workstation. The ultraviolet–visible spectroscopy (UV-Vis) absorption spectra were recorded using a Lambda 25 UV-Vis Spectrometer (PerkinElmer Lambda, Waltham, MA, USA). The Quanta FEG 250 scanning electron microscope (SEM; FEI, Tokyo, Japan) and the Jem-2100F transmission electron microscope (TEM; Jeol, Tokyo, Japan) were used to determine the morphology of the synthetized material. Energy dispersive spectroscopy (EDS) mapping using the Quanta FEG 250 (FEI, Tokyo, Japan) was used to conduct the elemental mapping of the prepared materials. Infrared spectra were recorded using a Nicolet iS50 Fourier transform infrared spectrometer (Thermo Fisher Scientific Inc., Waltham, MA, USA).

### 2.3. Synthesis of PEI-rGO-AuNCs Nanocomposites

#### 2.3.1. Preparation of AuNCs

The glutathione (GSH)-modified AuNCs were synthesized according to the reported literature [34]. Briefly, 5 mL HAuCl_4_ (5 mM) was mixed with 5 mL GSH (5 mM) in a 50 mL round-bottomed flask for 10 min until the solution was colorless. Then, it was heated to 80 °C. After reacting for 2 h in a water bath, the solution turned yellow. The yellow solution was cooled and filtered through a 0.22 μm membrane to remove any bulk gold.

#### 2.3.2. Preparation of PEI-rGO

The 20 mL of GO solution (0.2 mg/mL) was mixed with a solution of PEI (20.0 mL, 20 mg/mL), 1-(3-dimethylaminopropyl)-3-ethylcarbodiimide hydrochloride (EDC, 40 mg), and *N*-hydroxy succinimide (NHS, 48 mg). The reaction solution was stirred at room temperature for 12 h after adjusting the pH to about 7. Then, 60 μL of hydrazine monohydrate was added and stirred vigorously for 3 min, and the solution was heated at 95 °C for 1 h. The solution was centrifuged at 15,000× *g* for 15 min, discarding the supernatant and washing with deionized (DI) water. The above operation was repeated three times, and PEI-GO was then dispersed in 40 mL of DI water [35,36].

#### 2.3.3. Preparation of PEI-rGO-AuNCs

Excess GSH-modified AuNCs were added to PEI-rGO and sonicated for 3 min. Then, the mixture was centrifuged (3000× *g*, 5 min), and the supernatant was discarded. The precipitate was washed three times and resuspended in DI water (0.5 mg/mL) [33].

#### 2.3.4. Preparation of Electrochemical Sensor

Six microliters of the PEI-rGO-AuNCs was applied to cover the surface of the working electrode, which was placed in an oven at 60 °C for 15 min. Then, 3 μL of antibody solution (100 μg/mL) was dispensed onto the surface of the working electrode and stored at 4 °C for 12 h. Unreacted active sites were then blocked by adding 8 μL of 1% Bovine Serum Albumin (BSA) to the electrode surface and incubated for 45 min. The electrode was then rinsed with phosphate buffer solution (PBS, 0.1 M, pH = 7.0) three times and stored at 4 °C for further use.

#### 2.3.5. Electrochemical Assay of β-Lg

For the purpose of measuring β-Lg, 3-μL aliquots of β-Lg at different concentrations were pipetted onto the electrode and incubated at 25 °C for 45 min. The electrochemical sensors were then rinsed with PBS buffer (0.1 M, pH = 7.0) three times before the measurements. All the electrochemical measurements used cyclic voltammetry (CV) and differential pulse voltammetry (DPV) at room temperature and were performed in PBS containing 5.0 mM K_3_ (Fe(CN)_6_), 1.0 mM K_4_Fe(CN)_6_, and 0.1 mmol/L KCl. The parameters used for all the CV and DPV measurements are as follows: inter-electromotive force of −0.4 V, final electromotive force of 0.6 V, amplitude of 0.05 V, pulse width of 0.05 s, pulse period of 0.2 s, and scan rate of 0.05 V/s.

#### 2.3.6. Milk Sample Analysis

Four different brands of pure milk were screened for β-Lg: Jindian, Telunsu, Yili, and Mengniu (from a local store, China). The milk samples were mixed with deionized water at a dilution factor of 1:10,000 and centrifuged at 20,000× *g* for 15 min. The supernatant was collected and stored at −20 °C. The samples and five standard solutions (12.5, 25, 50, 100, and 200 μg/mL) were then analyzed with a β-Lg ELISA kit. For the purposes of detection by the electrochemical sensors, the milk samples were diluted before analysis. One hundred microliters of diluted extraction sample was diluted 10 times and incubated for 45 min at 25 °C with the sensors; DPV current responses were recorded. 

## 3. Results and Discussion

### 3.1. Electrochemical Immunoassay Strategy

A simple electrochemical sensor was developed with the PEI-rGO-AuNCs as the substrate platform. The manufacturing process of the sensor is shown in Figure 1. The amino groups in branched PEI and GO are used after the one-step reaction of the epoxy group to obtain PEI-GO, and PEI-rGO was obtained by chemical reduction with hydrazine monohydrate [35,36]. The water-soluble cationic PEI chain can improve the dispersibility and solubility of the material. The surface of PEI-rGO has a rich amino group (-NH_2_) with positive charge, which makes PEI-rGO compatible with other negatively charged nanomaterials [37,38]. The surface ligand of the AuNCs is glutathione, which makes the AuNCs carry a large number of negative charges. The GSH-stabilized AuNCs and PEI-rGO are assembled by electrostatic interaction. We used disposable screen-printed electrodes, which are divided into counter electrode (CE), reference electrode (RE), and working electrode (WE). The PEI-rGO-AuNCs composites are dripped onto the working electrode, which can amplify current signal [39,40,41], and the immune response between the antigen and antibody was used to observe the change in the current to analyze the β-Lg concentration. With this solution, rapid and highly sensitive detection of β-Lg can be achieved.

### 3.2. Characterization of PEI-rGO Nanocomposites

Fourier transform infrared spectroscopy (FTIR) confirmed the successful functionalization of rGO. Figure 2A shows the infrared spectra of GO and PEI-rGO. The spectrum of GO showed characteristic peaks of C=O (1722 cm^−1^), aromatic C=C (1579 cm^−1^), carboxyl OH (1399 cm^−1^), epoxide CO (1225 cm^−1^), and alkoxy C-O (1040 cm^−1^). Compared with GO, the characteristic peaks of oxide-containing groups in PEI-rGO disappeared, indicating that GO had been partially reduced during the reaction.

After being functionalized with PEI, a nucleophilic substitution reaction occurs between the epoxy group on GO and the terminal -NH_2_ on the PEI molecule, forming a new amide bond. Due to the presence of the nitrogen-containing surface group (-NH or -NH_2_) and trace H_2_O, a broad peak appeared at 3498 cm^−1^ [42]. The characteristic peak of the epoxy group at 1040 cm^−1^ weakened, and the characteristic peak of the CN group at 1456 cm^−1^ was generated. The amino group at the terminal of the PEI molecule was amidated with the carboxyl group on the GO so that the carbonyl peak at 1722 cm^−1^ disappeared, resulting in the vibration peak of amide group O=C-NH at 1613 cm^−1^, which indicated that PEI had been successfully covalently grafted to the rGO. Therefore, the sample after the covalent modification was termed PEI-rGO [43].

Figure 2B,C shows TEM images of GO and PEI-rGO. As shown in Figure 2B, the surface of GO is smooth and flat. When GO is reduced to rGO, the surface of the material will be wrinkled due to the poor water solubility of rGO. However, the water-soluble cationic PEI chain can improve the dispersibility and solubility of the material, and it can be seen from Figure 2C that the surface of PEI-rGO still exhibits good uniformity and flatness.

### 3.3. Characterization of PEI-rGO-AuNCs Nanocomposites

The UV-Vis absorption spectrum proved that the reduction of GO and the combination with PEI had occurred. GO generally has a distinct characteristic absorption peak at 230 nm, which is the π–π* transitional absorption of C=C of the aromatic ring. There is a weaker absorption peak at 300 nm, which is the n–π* transition of the C=O bond [44]. The conjugated electronic structure of graphene gradually recovered due to the reaction between GO and PEI and the reduction, and the maximum absorption peak of GO is obviously red-shifted. Comparing the spectrum of PEI-rGO-AuNCs with that of the AuNCs (Figure 3A), PEI-rGO-AuNCs has a weaker absorption peak at 270 nm, which matches the characteristic peak of PEI-rGO at 270 nm [44]. It can be seen that AuNCs have been successfully assembled on PEI-rGO due to the positive and negative charge attraction between AuNCs and PEI.

The morphology of PEI-rGO-AuNCs was characterized by TEM. It can be seen in Figure 3B that there are a large number of AuNCs with uniform particle size on the surface of PEI-rGO. Using EDS mapping to further explore the elements on the surface of the material, Au, C, and O can be clearly seen (Figure 3C,D).

### 3.4. Electrochemical Performance of the Electrochemical Sensors

Cyclic voltammetry (CV) is used to judge the reversibility, stability, and chemical reaction properties of electrodes [45,46]. The electrochemical behavior of different modified electrodes was evaluated by CV. As shown in Figure 4, according to the CV, the redox peak current of the electrode after modification with the composite was greater than that of the bare SPEs, which was attributed to the excellent conductivity of rGO and AuNCs. At the same time, the combination of PEI and rGO might provide the necessary conduction pathways on the electrode surface and a better electrochemical behavior. When the electrode surface was modified with the antibody, the peak current decreased; this was due to the antibody affecting the direct electron transfer on the electrode surface [45]. The above results indicated that the antigen was successfully immobilized on the electrode surface.

### 3.5. Optimization of Conditions for Electrochemical Analysis

The performance of an electrochemical sensor can be affected by different factors, such as the volume of material used to modify the electrode and the incubation time. In this study, experimental parameters were firstly optimized to obtain the best conditions. Figure 5A shows the current changes of PEI-rGO-AuNCs-modified electrode in the presence or absence of β-Lg. The results show that the response current reached 50 μA when the volume increased to 6 μL in the absence of β-Lg, after which the current declined. The current change of the electrode with and without β-Lg was calculated, as shown in Figure 5B. Finally, we chose to use the volume of 6 μL of PEI-rGO-AuNCs to detect β-Lg in subsequent experiments. The incubation time was investigated because it is crucial for recognizing β-Lg. As shown in Figure 5C,D, when the incubation time increased from 15 to 50 min, the current response and the rate of current change first increased and then stabilized at the maximum. Therefore, the best incubation time was 45 min.

### 3.6. Analytical Performance of β-Lg Detection

Under optimized conditions, the performance of the modified electrode was tested in different concentrations of β-Lg. Compared with the CV method, the DPV method has higher sensitivity when detecting an analyte at very low concentrations. Figure 6A shows the DPV current response of the electrochemical sensor, which gradually decreased with the increment of β-Lg. This can be ascribed to the electron transfer efficiency being affected by a large number of biomolecules. Figure 6B shows the calibration curve and linear range related to the logarithmic values of β-Lg concentration from 0.01 to 100 ng/mL. It can be seen that there is a good linear relationship between sensor and β-Lg concentration. The linear regression equation was Y = 11.396 + 2.126X, with a correlation coefficient (R^2^) of 0.992 and LOD of 0.08 ng/mL (S/*N* = 3). Compared with the purchased ELISA kit (LOD = 20 μg/mL), this electrochemical sensor had a comparable low detection limit and wide linear range.

### 3.7. Reproducibility, Specificity, and Stability of Sensor

The reproducibility is considered to be one of the most important factors in evaluating the precision of the sensor. As shown in Figure 7A, five replicate sensors for the detection of β-Lg were measured in PBS (pH 7.0). The relative standard deviation of repeated measurements was calculated as 1.9%, indicating the favorable reproducibility of the designed sensors.

The selectivity of proposed PEI-rGO-AuNCs sensor was used to evaluate the feasibility of β-Lg detection in complex systems. Four kinds of proteins, namely egg albumin, BSA, egg lysozyme, and casein, were considered as diverse interfering substances in the food for the test. As illustrated in Figure 7B, interfering substances did not exhibit significant changes of current signal, and relevant variations were less than 4 μA. However, great changes of current (12 μA) were observed in the presence of β-Lg. The results indicated that the designed sensor had excellent selectivity and specificity for β-Lg detection.

In addition, the stability is also an essential parameter in evaluating the performance of a sensor. To confirm the long-term storage stability of this sensor, three modified electrodes were stored in a refrigerator at 4 °C for two weeks and tested in the same testing condition. As shown in Figure 7C, the current signals were only shifted by 0.9, 3.2, and 4.3%, confirming that the designed electrochemical sensors had good stability. All of these results verified that the sensor designed in this work possessed potential practical value.

### 3.8. Milk Sample Analysis

In order to test the detection ability of a modified electrode in real samples, the developed electrochemical sensors were then used to detect β-Lg in four brands of pure milk commonly found in the marketplace. As shown in Table 1, the measurement results of electrochemical sensors and ELISA are relatively similar, which indicates that the sensors can be conventionally used for the measurement of β-Lg. In addition, the detection limit of the ELISA is at least 20 μg/mL, while the detection limit of the electrochemical sensors is lower than that of the ELISA.

## 4. Conclusions

Sensitive electrochemical sensors based on AuNCs and graphene-modified screen-printed electrodes for detecting β-Lg have been developed. The method of synthesizing the sensing material for the modified electrode is simple and low-cost. The electrochemical sensor has been successfully applied to detect β-Lg in samples, and the results are similar to those of the ELISA methods that provide reliable analytical screening tools. The sensor enabled a limit of detection (LOD) of 0.08 ng/mL and a detection range from 0.01 to 100 ng/mL of β-Lg. Therefore, the sensor prepared in this paper is small in size and easy to carry, and it is expected to be used in the manufacture of portable detection equipment.

## Figures and Tables

**Figure 1 sensors-20-03956-f001:**
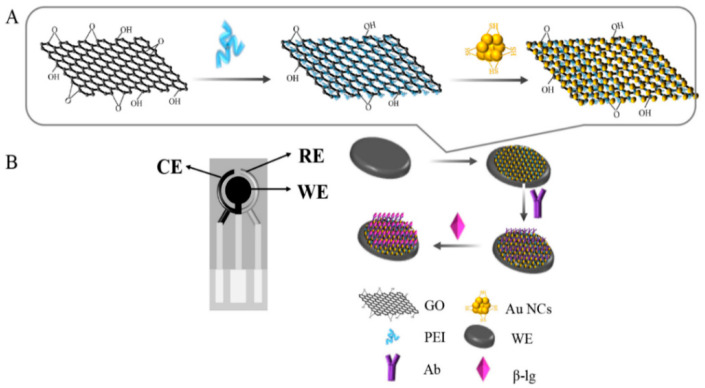
Basic principles of this research strategy: (**A**) assembly process of the composite material obtained by modifying reduced graphene oxide (rGO) with polyethyleneimine (PEI) and gold nanoclusters (AuNCs) (PEI-rGO-AuNCs); (**B**) assembly steps of electrochemical sensors. CE = counter electrode, RE = counter electrode, and WE = working electrode.

**Figure 2 sensors-20-03956-f002:**
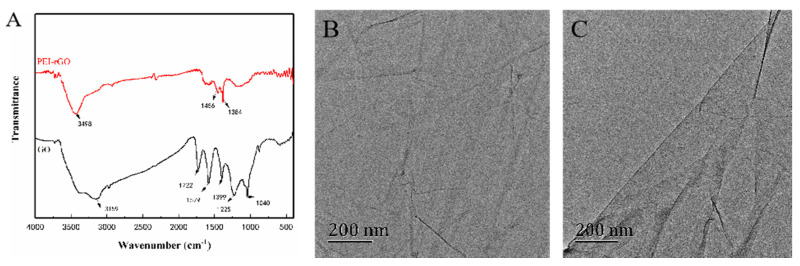
(**A**) FTIR spectra of the GO and the PEI-rGO. TEM images of (**B**) the GO and (**C**) the PEI-rGO.

**Figure 3 sensors-20-03956-f003:**
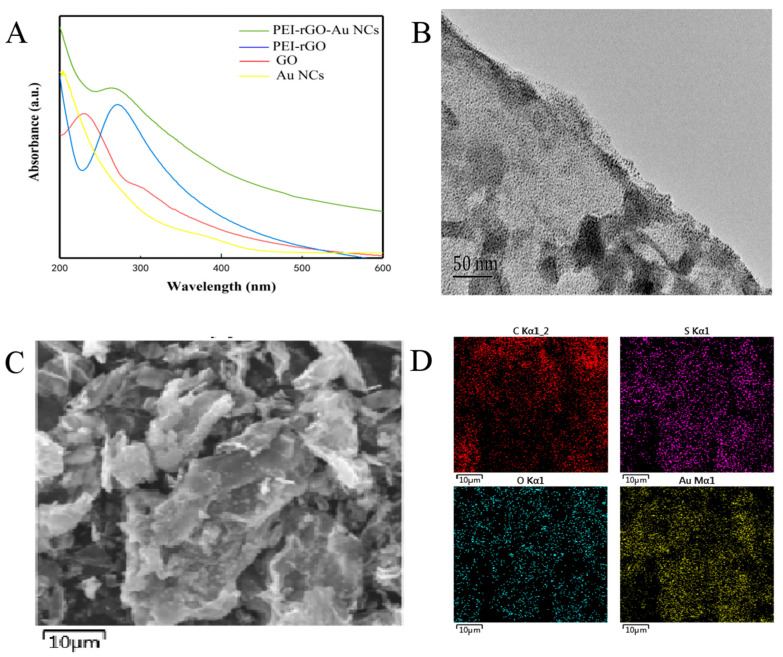
(**A**) UV-Vis spectra of PEI-rGO-AuNCs, PEI-rGO, GO, and AuNCs in the wavelength range from 300 to 600 nm; (**B**) TEM image of PEI-rGO-AuNCs assembly; (**C**) SEM image of PEI-rGO-AuNCs assembly; (**D**) EDS of PEI-rGO-AuNCs confirming the presence of elemental C, S, O, and Au.

**Figure 4 sensors-20-03956-f004:**
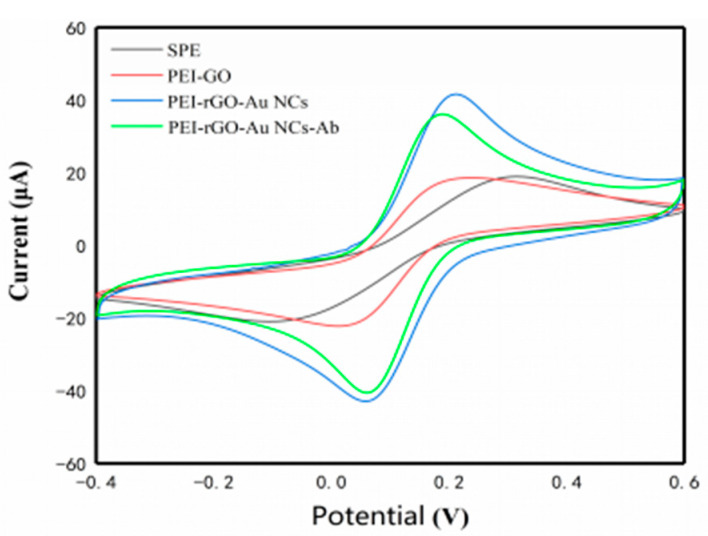
Characterization and analytical performance of the electrochemical sensor. Typical CV studies of the SPE, PEI-rGO (6 μL, 0.5 mg/mL), PEI-rGO-AuNCs (6 μL, 0.5 mg/mL), and PEI-rGO-AuNCs–antibody (3 μL, 100 μg/mL)-modified electrode in 5.0 mM K_3_ (Fe(CN)_6_), 1.0 mM K_4_Fe(CN)_6_, and 0.1 mM KCl.

**Figure 5 sensors-20-03956-f005:**
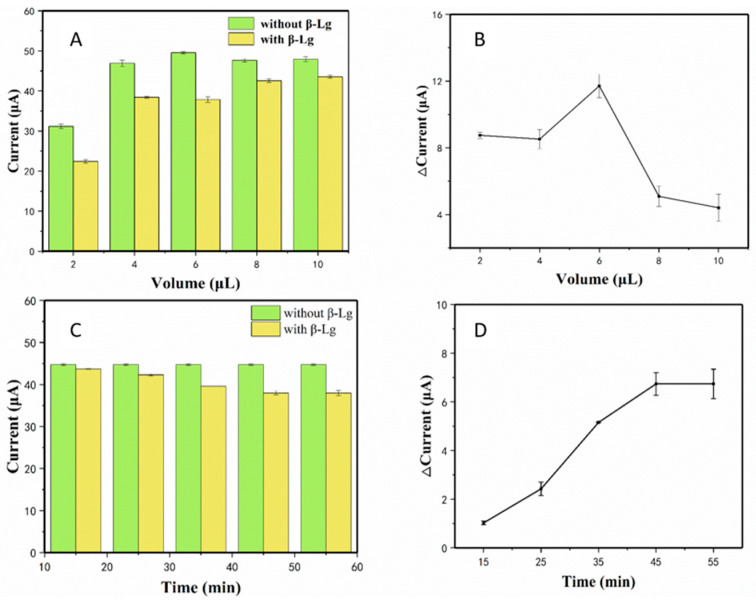
Optimization conditions for β-lactoglobulin (β-Lg) detection using the electrochemical assay: (**A**,**B**) the titration volume (2, 4, 6, 8, or 10 μL) of PEI-rGO-AuNCs used to modify the electrode; (**C**,**D**) the incubation time (10, 15, 25, 35, 45, or 55 min) of the PEI-rGO-AuNCs electrode with β-Lg standards (0.025 ng/mL).

**Figure 6 sensors-20-03956-f006:**
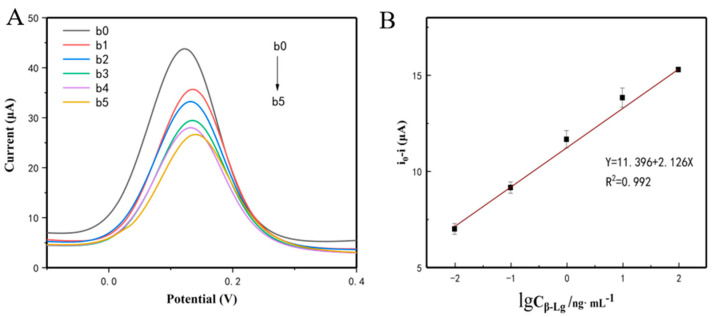
(**A**) Differential pulse voltammetry (DPV) responses of the proposed electrochemical sensors after incubation with different concentrations of β-Lg (b0 = 0 ng/mL, b1 = 0.01 ng/mL, b2 = 0.1 ng/mL, b3 = 1 ng/mL, b4 = 10 ng/mL, and b5 = 100 ng/mL). (**B**) The calibration curve of the developed electrochemical sensors for β-Lg.

**Figure 7 sensors-20-03956-f007:**
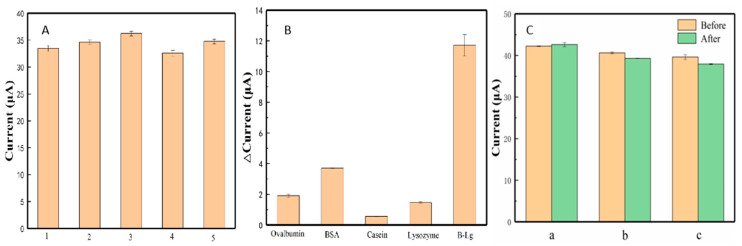
Results of (**A**) reproducibility, (**B**) specificity (0.025 ng/mL β-Lg, ovalbumin, BSA, casein, lysozyme), and (**C**) stability study on the electrochemical sensor.

**Table 1 sensors-20-03956-t001:** Comparison of β-Lg detection in real samples using the electrochemical sensors and ELISA.

Sample	ELISA (μg/mL)	Sensor (μg/mL)
Jindian	10.38 ± 0.02	10.85 ± 0.25
Telunsu	17.09 ± 0.56	17.05 ± 0.55
Yili	15.54 ± 0.47	16.10 ± 0.10
Mengniu	15.47 ± 0.40	14.60 ± 0.60

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
