# Peer review of "An Electrochemical Sensor Based on Gold-Nanocluster-Modified Graphene Screen-Printed Electrodes for the Detection of β-Lactoglobulin in Milk"

_sensors, 2020, doi:10.3390/s20143956_

Round 1
Reviewer 1 Report
Please, find the comments in the attached PDF.

Author Response
Response to Reviewer 1 Comments Point 1: In Fig. 3A the optical absorbance of the different compounds are plotted vs wavelength. The AuNCs spectrum has been shown as an inset to the figure and this is very small, blurry and not readable. Please improve it. Response 1: We thank the reviewer’s constructive suggestion. According to the reviewer’s suggestion, the optical absorbance of AuNCs was removed from the inset figure into figure 3A, which is shown in Line Point 2: In Fig. 6A the sensor response is shown for different concentrations of β-Lg. However the concentrations are simply expressed as b0, b1, ….., b5. It should be better to show the values of the concentrations. This can also be done by expressing the concentration values of b0, b1, ….., b5 in the figure caption. Response 2: Following the reviewer’s suggestion, we have added the concentrations of β-Lg from b0 to b5 in the title of Fig. 6A in Line 247-248, page 9 in the revised version manuscript, and also deleted the corresponding description in the 3.6 section of the text. Point 3: Please, correct some typos present in the text such as: at line 116 “applid” should be “applied”; at line 149 “can amplified” should be “can amplify; at line 217 “to detection” should be “to detect”. Response 3: We thank the reviewer for pointing out the catachresis above. According to the reviewer’s suggestion, we have corrected the “applid” into “applied” in Line 122, page 3; the “can amplified” into “can amplify” in Line 157, page 4; the “to detection” into “to detect” in Line 225, page 7 in the revised version manuscript.

Reviewer 2 Report
The paper describes the development a biosensor using a support formed by gold nanocluster/graphene to immobilize an antibody on carbon-printed electrode to determine β-lactoglobulin on milk samples.
The manuscript presents interesting results, however in my opinion, it is necessary to improve some information before publication. Some suggestions are presented below:
- In the introduction the authors do not explain which the advantages to use a gold nanocluster. This component is part of the title and, therefore, its use should be better explained in the text. Additionally, the authors could be describe if there are other works in literature that present the application of the composite (polyethyleneimine (PEI)-reduced graphene oxide (rGO)-gold nanoclusters) for biosensor development.
- There are some components that do not be described in the manuscript, for example: GSH and the kind of antibody used.
- In Figure 1 some parts are difficult to visualize (functional groups, for example). I suggest improving the resolution of the Figure.
- To highlight the advantages of this sensor in relation to similar ones in the literature, it is essential to have a comparative table showing some figures of merit. There are studies that present better detection limits, such as:
Biosensors and Bioelectronics, Volume 38, Pages 308-313, 2012.
Analytica Chimica Acta, Volume 1120, Pages 1-10, 2020.
- The captions of the figures should be more complete, with antigen concentration, pulse amplitude, scan rate.
- The magnitude of current shown in Figures 5 and 7 is the same that b1 in the calibration curve, but the authors described the use of 25 ng/L β-Lg solution. This point is necessary to revise.
- The novelty of the dispositive is not well presented in the manuscript.
Author Response
Response to Reviewer 2 Comments
Point 1: In the introduction the authors do not explain which the advantages to use a gold nanocluster. This component is part of the title and, therefore, its use should be better explained in the text. Additionally, the authors could be describe if there are other works in literature that present the application of the composite (polyethyleneimine (PEI)-reduced graphene oxide (rGO)-gold nanoclusters) for biosensor development.
Response 1: We very much appreciate the reviewer’s positive comments. According to the reviewer’s suggestion, we have supplemented the advantages and applications of gold nanoclusters in Line 65-68, page 2 in the revised version manuscript. We make the description of metal nanoparticles more clear, which refers directly to gold nanoclusters in the text. The corresponding references were also changed, which was shown in Line 71, page 2 in the revised version manuscript. Furthermore, the composite of PEI-rGO-AuNCs has been once developed to detect nonylphenol, which was presented in the Introduction section in Line 77, page 2 in the revised version manuscript.
Point 2: There are some components that do not be described in the manuscript, for example: GSH and the kind of antibody used.
Response 2: We thank the reviewer for pointing out the mistakes above. According to the reviewer’s suggestion, we added the description of GSH and the kind of antibody, we changed “GSH” into “glutathione (GSH)” in Line 104, page3 and “Antibody” into “Anti-β-lactoglobulin antibody” in Line 87, page 2 in the revised version manuscript. Furthermore, other similar mistakes were changed correspondingly in Line 111-112, page 3, Line 115, page 3, Line 164, page 5 in the revised version manuscript.
Point 3: In Figure 1 some parts are difficult to visualize (functional groups, for example). I suggest improving the resolution of the Figure.
Response 3: Following to the reviewer’s suggestion, we have redrew Figure 1 with the improved image resolution to ensure that the function group is clearly visible, which can be seen in Line 160, page 4 in the revised version manuscript.
Point 4:The captions of the figures should be more complete, with antigen concentration, pulse amplitude, scan rate.
Response 4: According to the reviewer’s suggestion, we have revised the corresponding description of captions of Figure 1, 3, 4, 5, 6, 7 in order to making the experimental procedure more detailed, which can be found in the revised manuscript. Also, we added “Scan rate: 0.05 V/S” in Line 135, page 3, the pulse amplitude and other parameters were presented in Line 134-135, page 3 in the revised version manuscript. All the CV and DPV measurements were under these parameters.
Point 5:The magnitude of current shown in Figures 5 and 7 is the same that b1 in the calibration curve, but the authors described the use of 25 ng/L β-Lg solution. This point is necessary to revise.
Response 5: Thank you for the reviewer’s advice. It was our mistake not to make the unit of concentration consistent, we have changed “25 ng/L” into 0.025 ng/mL in Line 233, page 8, and Line 268, page 9 in the revised version manuscript.
Point 6:To highlight the advantages of this sensor in relation to similar ones in the literature, it is essential to have a comparative table showing some figures of merit. There are studies that present better detection limits, such as:
Biosensors and Bioelectronics, Volume 38, Pages 308-313, 2012.
Analytica Chimica Acta, Volume 1120, Pages 1-10, 2020.
The novelty of the dispositive is not well presented in the manuscript.
Response 6: Thank you for the reviewer’s advice. Firstly, compared with our previous research (Xu et al. Analytica Chimica Acta. 2020, 1120, 1-10.), we used screen-printed electrodes instead of the traditional three-electrode system, without cumbersome electrode grinding and polishing steps. In contrast with Eissa’s study
(Eissa et al. Biosens. Bioelectron. 2012, 38, 308-313.), the modification of electrodes was simple dropcoating method, which was based on the self-assembly of PEI-rGO and AuNCs. Secondly, In recent years, in order to make equipment for field detection, sensors need to have small and portable size, simple operation, fast response speed, and moderate detection limit required by the local government. Therefore, the novelty of our study is emphasized and compared as follows “Compared with our previous strategy using an ultra-highly selective DNA aptamer and the modified screen-printed carbon electrode by choosing complicated grafting protocol [20, 32], the major objective of this study is to establish an electrochemical sensor with simple operation, low cost, fast response and moderate detection limit required by the local government, which is expected to be applied to portable field rapid detection equipment for the detection of β-Lg.”, which was presented in Line 80-84, page 2 in the revised version manuscript.
References:
Xu, S.; Dai, B.; Zhao, W.; Jiang, L.; Huang, H. (2020). Electrochemical detection of β-lactoglobulin based on a highly selective DNA aptamer and flower-like Au@ BiVO4 microspheres. Analytica Chimica Acta. 2020, 1120, 1-10.
Eissa, S.; Tlili, C.; L"Hocine, L.; Zourob, M. Electrochemical immunosensor for the milk allergen β-Lg based on electrografting of organic film on graphene modified screen-printed carbon electrode. Biosens. Bioelectron. 2012, 38, 308-313.

Round 2
Reviewer 2 Report
The manuscript was corrected according to suggestions.